# A general algorithm for computing bound states in infinite tight-binding systems

Mathieu Istas[1*], Christoph Groth[1], Anton R. Akhmerov[3], Michael Wimmer[2, 3] and Xavier Waintal[1]

**1** Univ. Grenoble Alpes, INAC-PHELIQS, F-38000 Grenoble, France
**2** QuTech, Delft University of Technology, 2600 GA Delft, The Netherlands
**3** Kavli Institute of Nanoscience, Delft University of Technology,
P.O. Box 4056, 2600 GA Delft, The Netherlands

* mathieu.istas@cea.fr

## Abstract

We propose a robust and efficient algorithm for computing bound states of infinite tight-binding systems that are made up of a finite scattering region connected to semi-infinite leads. Our method uses wave matching in close analogy to the approaches used to obtain propagating states and scattering matrices. We show that our algorithm is robust in presence of slowly decaying bound states where a diagonalization of a finite system would fail. It also allows to calculate the bound states that can be present in the middle of a continuous spectrum. We apply our technique to quantum billiards and the following topological materials: Majorana states in 1D superconducting nanowires, edge states in the 2D quantum spin Hall phase, and Fermi arcs in 3D Weyl semimetals.

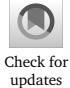
# 1 Introduction

Simulating quantum devices that are connected to infinite leads is a commonly occurring problem in quantum nanoelectronics. In particular, finding the propagating states in the leads and coupling them to the device allows to evaluate transport properties such as the conductance of the device [1]. Numerical methods for solving this scattering problem have a long history [2, 3, 5]. Modern stable algorithms are available in several software packages, e.g. Refs. 6–8.

The focus of this work is identifying bound states: states, that are localized inside the central region and decay exponentially in the leads or any other translationally invariant bulk of the material (see Fig. 1). Although these states do not contribute to transport, they frequently are the central object of study. Examples of bound states relevant to current research include the quantum well states in semiconductor heterostructures, surface states in metals, impurity states (phosphorus donors in silicon, nitrogen-vacancy centers in diamond [9], Shiba states due to magnetic impurities in superconductors), Andreev states [10] in Josephson junctions, and various kinds of protected edge states in topological materials (e.g. Majorana states in superconducting nanowires [11–13], chiral edge states in quantum spin Hall systems or Fermi arcs in Weyl semi-metals). Computing the bound states can also be desirable from a mathematical or technical perspective: the calculations of Feynman diagrams due to electron-electron interactions requires the full basis of the system (see for instance Ref. 14).

Closed-form solutions of the bound state problem exist for several regimes, such as the strong coupling (or short junction) limit of superconducting junctions [15, 16], or the semi-classical approach to impurity bound states [17].

To solve that problem, one could also use a brute-force method, by cutting out a finite part of the infinite system and diagonalizing the corresponding sub-Hamiltonian. A more evolved numerical method using the boundary element method has been developed in Ref 18, but its drawback is that the scattering region must consist of several homogeneous region and only compute the bound states at negative energies, while our algorithm does not bear these constraints. If the state decays fast enough in the leads and a sufficiently large portion of them has been included, this results in a precise determination of the bound states. This approach is however not always satisfactory due to its significant computational overhead when the decay length of the bound state diverges. In addition, this brute-force method by itself does

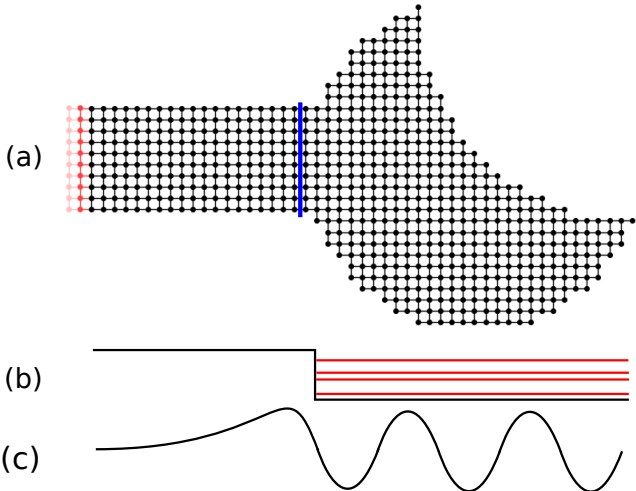

Figure 1: Panel (a): an example system considered in this work consisting of a scattering region (right of the blue line) connected to a semi-infinite lead (on the left, the red cells are to be repeated up to infinity). (b): The energy spectrum shows several discrete levels next to the continuous spectrum of the bands. (c): A schematic representation of a bound state wavefunction, decaying exponentially fast in the lead.

not allow to distinguish the bound states from the continuum spectrum when an exact bound state coexists with the continuum spectrum [19]. We present an algorithm for calculating the bound state spectrum directly for the infinite system. Compared to the brute force method, our approach has the following advantages:

- It is typically more efficient because it operates with smaller matrices than a truncated finite system.

- It provides an exact asymptotic form of the bound states inside the lead.

- Its performance is not hindered by the presence of slowly decaying modes.

- It allows to reliably distinguish bound states from scattering states including the situation when the scattering states exist at the same energy as a bound state.

The outline of this article is as follows. Sec. 2 introduces the generic model and the formulation of the bound state problem. The algorithm is developed is Sec. 3. The first application, a quantum non-homogeneous discretized billiard, as opposed to continuous ones like in Ref. 20, is presented in Sec. 4 where we study the difference between integrable and chaotic billiards and show that in the former there exist bound states in the continuum (BICs) as in Ref. 21. (Indeed, the ability of our algorithm to isolate bound states from the continuum is one of its strengths.) Sections 5.1, 5.2, and 5.3 continue with further applications: the calculation of edge states for three different kinds of topological phases. These are, respectively, a 1D Majorana bound state in a superconducting wire, a 2D quantum spin Hall phase within the Bernevig-Hughes-Zhang (BHZ) model, and Fermi arcs in a 3D Weyl semimetal.

## 2 The bound state problem

### 2.1 General model

A typical system of interest is sketched in Fig. 1: a central system of $N_\text{sr}$ orbitals is connected to one or more leads. The leads themselves are semi-infinite and periodic. They consist of unit cells that contain $N_\text{t}$ orbitals each and are repeated up to infinity. Without loss of generality, we can assume that there is only one lead: if there are several, they all can be considered as a single effective lead that is made up of disconnected parts. To simplify the notation, we include the first unit cell of the lead in the scattering region. With these conventions, the total Hamiltonian of the system can be written as

$$\hat{H}_\text{tot} = \begin{pmatrix} H_\text{sr} & P_\text{sr}^T V^\dagger & & & \\ V P_\text{sr} & H & V^\dagger & & \\ & V & H & V^\dagger & \\ & & V & H & \ddots \\ & & & \ddots & \ddots \end{pmatrix}, \tag{1}$$

where the Hamiltonian of the scattering region is a finite but potentially big $N_\text{sr} \times N_\text{sr}$ matrix $H_\text{sr}$, the Hamiltonian $H$ acting within each unit cell of the lead is a $N_\text{t} \times N_\text{t}$ matrix, while the $N_\text{t} \times N_\text{t}$ hopping matrix $V$ describes the coupling of neighboring unit cells. The $N_\text{t} \times N_\text{sr}$ diagonal rectangular matrix $P_\text{sr}$ is defined as $[P_\text{sr}]_{ij} = 1$ when the site $i$ of the first unit cell of the lead is coupled to the site $j$ of the scattering region and zero otherwise. In the case where the system has several leads, the matrices $H$ and $V$ are block-diagonal with each block corresponding to a single lead.

We search for the eigenstates $\hat{\psi}$ of the matrix $\hat{H}_\text{tot}$, i.e. the solutions of

$$\hat{H}_\text{tot} \hat{\psi} = E \hat{\psi}.$$

For finite-sized problems, this amounts to diagonalizing a finite Hermitian matrix. For infinite systems, however, two cases arise depending on whether the energy $E$ lies within one of the bands of the lead or whether it corresponds to a localized eigenstate. The first case—the scattering problem—has been extensively studied in the literature in various formulations, see Refs. 2, 22–24, and can be cast into solving a set of linear equations [6]. Because the scattering problem has a full set of solutions for any $E$ belonging to the continuum spectrum of the lead, the energy $E$ is an input of the calculation. In contrast, in the second case—the bound state problem—the energy is an output of the calculation since localized eigenstates exist only for specific values of $E$.

### 2.2 Lead modes

In the spirit of the scattering problem let us first analyze the possible modes: states that exist in the lead at a given energy. Since the lead is translationally invariant, any wave function can be decomposed into the eigenstates of the translation operator,

$$\phi(j) = \lambda^j \phi, \tag{2}$$

where $\lambda = e^{ik}$ characterizes the momentum in the lead and $j$ labels its unit cells (the index grows with increasing distance from the scattering region). The Schrödinger equation in the lead takes the form

$$V\phi + \lambda(H - E)\phi + \lambda^2 V^\dagger \phi = 0 \tag{3}$$

or alternatively

$$H(k)\phi = E\phi,$$

where we have introduced the Bloch Hamiltonian

$$H(k) = H + Ve^{-ik} + V^{\dagger}e^{ik}. \tag{4}$$

Introducing $\xi \equiv \lambda\phi$ we transform the Eq. (3) the generalized eigenstate problem

$$\begin{pmatrix} H-E & V^{\dagger} \\ 1 & 0 \end{pmatrix} \begin{pmatrix} \phi \\ \xi \end{pmatrix} = \frac{1}{\lambda} \begin{pmatrix} -V & 0 \\ 0 & 1 \end{pmatrix} \begin{pmatrix} \phi \\ \xi \end{pmatrix}, \tag{5}$$

that can, be solved using standard linear algebra routines (in practice this step requires some care and will be discussed extensively in Ref. 25). Solutions of the Eq. (5) with $|\lambda| = 1$ (real values of the momentum $k$) are propagating while those with $|\lambda| < 1$ are evanescent. Finally, modes with $|\lambda| > 1$ diverge exponentially with the distance from the scattering region and therefore they play no role in the bound state problem.

## 2.3 Formulation of the bound state problem

Inside the lead an eigenstate $\hat{\psi}$ can be expressed as a superposition of propagating and evanescent modes. We can now formulate the bound state problem as follows: *a bound state is an eigenstate $\hat{\psi}$ that has no overlap with the propagating modes, i.e. that is purely evanescent.* To be more specific, we gather all the eigenvalues $|\lambda| < 1$ at a given energy $E$ into a diagonal matrix $\Lambda_e(E)$ (of size $N_e \times N_e$) and the corresponding evanescent eigenstates $\phi$ into the matrix $\Phi_e(E)$ (each column of the $N_t \times N_e$ matrix $\Phi_e$ is a vector $\phi$ corresponding to an evanescent state). With these definitions Eq. (3) takes the form

$$V\Phi_e + (H-E)\Phi_e\Lambda_e + V^{\dagger}\Phi_e(\Lambda_e)^2 = 0. \tag{6}$$

Using this notation, an eigenstate $\hat{\psi}$ assumes the following general form in the lead:

$$\psi(j > 0) = \Phi_e(\Lambda_e)^j q_e, \tag{7}$$

where the vector $q_e$ contains the coefficients of the expansion in terms of the evanescent states. Denoting the subvector of $\hat{\psi}$ inside the scattering region $\psi_{sr}$, we arrive at the following equation with unknown $(\psi_{sr}, q_e)$:

$$\begin{pmatrix} H_{sr}-E & P_{sr}^T V^{\dagger}\Phi_e(E)\Lambda_e(E) \\ VP_{sr} & -V\Phi_e(E) \end{pmatrix} \begin{pmatrix} \psi_{sr} \\ q_e \end{pmatrix} = 0. \tag{8}$$

Here we have reinstated the explicit energy dependence of the mode matrices, and this result is very similar to Eq. (D2) of Ref. 4, but the authors did not look for a way to formulate this nonlinear problem in an efficient and practical algorithm. We therefore reduce the bound state problem to finding $E$, $\psi_{sr}$, and $q_e$ satisfying the Eq. (8).

Eliminating $q_e$ from the Eq. (8) yields an equivalent equation

$$[H_{sr} + \Sigma(E)]\psi_{sr} = E\psi_{sr}, \tag{9}$$

where the self-energy $\Sigma$ is

$$\Sigma(E) = P_{sr}^T V^{\dagger}\Phi_e\Lambda_e\frac{1}{V\Phi_e}VP_{sr}. \tag{10}$$

In the remainder, we focus on the formulation Eq. (8) and do not make use of the alternative self-energy formulation. Note that Eq. (10) is only defined when the matrix $V$ is invertible. A better definition of the self-energy is given in the next section.

# 3 Numerical algorithm for finding bound states

We now turn to the construction of practical algorithms that use Eq. (8) to calculate the bound states of an infinite system. For completeness and pedagogical purposes, we present three different algorithms of increasing effectiveness. Algorithm III is numerically superior to the other two.

## 3.1 Algorithm I: non-Hermitian formulation

The first algorithm is formulated directly in the non-Hermitian representation of Eq. (8). The first thing to notice is that when the matrix $V$ is not invertible, it admits trivial solutions. As an example, let us consider

$$V = \begin{pmatrix} 0 & 1 \\ 0 & 0 \end{pmatrix}.$$

One immediately sees that this leads to the last row of Eq. (8) to be made up of zeros, so that the equation admits solutions for any energy $E$. However, while one can find a non-zero vector $q_e$, it corresponds to a vanishing eigenvalue $\lambda$ for a vector $\phi$ that belongs to the kernel of $V$ (i.e. $V\phi = 0$) so that the actual state given by Eq. (7) vanishes everywhere and is therefore not a true solution.

To remove these spurious solutions, we perform a singular value decomposition of $V$ that we write as $V = U_V D W_V^\dagger$, where $U_V$ and $W_V$ are unitary matrices while $D$ is diagonal. We order the matrices $D$ and $\Lambda_e$ such that their vanishing eigenvalues are placed in the lower right part of the matrix. It can be noted that the number of eigenvalues $\lambda = 0$ is equal to the dimension of the kernel of $V$, as seen form Eq. (3). Discarding the zero-valued trailing rows of Eq. (8) and an equal number of columns, that are either made of zeros or correspond to modes associated with $\lambda = 0$ contributions, we arrive at

$$\begin{pmatrix} H_{sr} - E & P_{sr}^T V^\dagger \tilde{\Phi}_e \tilde{\Lambda}_e \\ \tilde{D} \tilde{W}_V^\dagger P_{sr} & -\tilde{D} \tilde{W}_V^\dagger \tilde{\Phi}_e \end{pmatrix} \begin{pmatrix} \psi_{sr} \\ \tilde{q}_e \end{pmatrix} = 0, \tag{11}$$

where $\tilde{\Phi}_e$, $\tilde{\Lambda}_e$, $\tilde{q}_e$, $\tilde{D}$ and $\tilde{W}_V^\dagger$ are the truncated versions of $\Phi_e$, $\Lambda_e$, $q_e$, $D$ and $W_V^\dagger$, where the elements that play no role are removed. The matrix in Eq. (11) has the size $(N_{sr}+N_V) \times (N_{sr}+N_e)$, where $N_V$ is the rank of $V$. Note that the matrix is square if and only if there are no propagating modes.

The bound state problem is now reduced to finding the values of $E$ for which the left-hand side $Q(E)$ of Eq. (11) is not invertible. However, $Q(E)$ is in general not Hermitian and not even a square matrix. Singular value decomposition provides a solution to this problem: we form the matrix $Q^\dagger(E)Q(E)$ and obtain its lowest eigenvalues using a sparse solver (we use the Arpack implementation of the Lanczos algorithm using the shift-invert technique). Since the matrix $Q^\dagger Q$ is positive definite its eigenvalues are also positive (see an example in Fig. 2b), we are hence looking for zero singular values usind standard one-dimensional minimization algorithms. They are however less efficient than the root-finding algorithms that we apply in algorithms II and III.

Algorithm I is similar to the algorithm introduced in Refs. 26, 27 which maps the bound state problem on the solution of a non-linear eigenvalue problem of a non-Hermitian matrix. There is, however, a notable difference: Refs. 26, 27 looks for a vanishing determinant instead of the lowest singular value. This is less numerically efficient since determinant calculations can easily overflow/loose precision for large matrices and they do not exploit the matrix sparsity structure.

## 3.2 Hermitian formulation

To improve on algorithm I, we slightly reformulate Eq. (8) which is in general overcomplete since the matrix on the left-hand side is $(N_{sr} + N_t) \times (N_{sr} + N_e)$. By multiplying the second line of Eq. (8) by $\Lambda_e^* \Phi_e^\dagger$ we obtain a set of *necessary* conditions for a bound state to occur:

$$\begin{pmatrix} H_{sr} - E & P_{sr}^T V^\dagger \Phi_e \Lambda_e \\ \Lambda_e^* \Phi_e^\dagger V P_{sr} & -\Lambda_e^* \Phi_e^\dagger V \Phi_e \end{pmatrix} \begin{pmatrix} \psi_{sr} \\ q_e \end{pmatrix} = 0, \tag{12}$$

with the explicit dependence on energy $E$ removed for clarity. The matrix on the left-hand side of Eq. (12) now has a square $(N_{sr} + N_e) \times (N_{sr} + N_e)$ shape and is moreover Hermitian, a desirable property for numerical purposes. Indeed, as we show in App. B, Eq. (6) leads to

$$\Lambda_e^* \Phi_e^\dagger V \Phi_e = \Phi_e^\dagger V^\dagger \Phi_e \Lambda_e, \tag{13}$$

and therefore to Hermiticity of Eq. (12).

The next step is to get rid of the spurious $\lambda = 0$ solutions that can be present in $\Lambda_e$. We reorder the $\Lambda_e$ matrices so that their vanishing eigenvalues are placed in the lower right part of the matrix. After this reordering the corresponding last rows of Eq. (12) vanish, and we therefore remove them from the matrix. Similarly, by using Eq. (13), we also remove the last columns of the matrix. Introducing truncated quantities $\tilde{\Phi}_e$, $\tilde{\Lambda}_e$, and $\tilde{q}_e$ where the zero entries due to the $\lambda = 0$ contributions have been disregarded, we finally arrive at

$$\begin{pmatrix} H_{sr} - E & P_{sr}^T V^\dagger \tilde{\Phi}_e(E) \tilde{\Lambda}_e(E) \\ \tilde{\Lambda}_e^*(E) \tilde{\Phi}_e^\dagger(E) V P_{sr} & -\tilde{\Lambda}_e^*(E) \tilde{\Phi}_e^\dagger(E) V \tilde{\Phi}_e(E) \end{pmatrix} \begin{pmatrix} \psi_{sr} \\ \tilde{q}_e \end{pmatrix} = 0, \tag{14}$$

the central result of this article, where we show the explicit energy dependence to emphasize the non-linearity of the eigenproblem.

Eliminating $\tilde{q}_e$ from the Eq. (14) we also obtain an alternative expression of the self-energy that is defined even when $V$ is not invertible:

$$\Sigma(E) = P_{sr}^T V^\dagger \tilde{\Phi}_e \tilde{\Lambda}_e \frac{1}{\tilde{\Lambda}_e^* \tilde{\Phi}_e^\dagger V \tilde{\Phi}_e} \tilde{\Lambda}_e^* \tilde{\Phi}_e^\dagger V P_{sr}. \tag{15}$$

Combining this expression with Eq. (9) provides an equivalent alternative to the algorithms presented below. We do not pursue this route because there is no a-priori indication of any benefits from using the self-energy formulation, either by solving directly Eq. (9) or using an iterative scheme.

## 3.3 Algorithm II: root finder in the Hermitian formulation

The problem is now set in a Hermitian form suitable for numerical computation. Let us abbreviate Eq. (14) as

$$H_{eff}(E) \psi_{eff} = 0 \tag{16}$$

with

$$H_{eff} \equiv \begin{pmatrix} H_{sr} - E & P_{sr}^T V^\dagger \tilde{\Phi}_e \tilde{\Lambda}_e \\ \tilde{\Lambda}_e^* \tilde{\Phi}_e^\dagger V P_{sr} & -\tilde{\Lambda}_e^* \tilde{\Phi}_e^\dagger V \tilde{\Phi}_e \end{pmatrix}, \tag{17}$$

and

$$\psi_{eff} \equiv \begin{pmatrix} \psi_{sr} \\ \tilde{q}_e \end{pmatrix}. \tag{18}$$

The $E$-dependence has not been written explicitly in Eq. (17), but the solutions $\tilde{\Lambda}_e(E)$ and $\tilde{\Phi}_e(E)$ of Eq. (6) are nontrivial functions of $E$, which makes the eigenproblem a nonlinear

one. We are interested in finding the values of $E$ and the corresponding eigenvectors at which Eq. (16) admits nontrivial solutions. A necessary condition for this is the presence of a zero eigenvalue of $H_{\text{eff}}(E)$. For any given $E$, the eigenvalues of $H_{\text{eff}}(E)$ that are close to zero can be computed using a sparse solver (again, we use the Arpack implementation of the Lanczos algorithm using the shift-invert technique). The values of $E$ where any eigenvalue vanishes can be then found using one of the many standard root-finding algorithms for one-dimensional functions.

Once a vanishing eigenvalue has been found, a check is necessary in order to avoid a solution that is not a physical bound state. A trivial example of such a false positive is an eigenstate of $H_{\text{sr}}$ at an energy $E$ such that there are no evanescent states. In that case the matrices $\Phi_e$ and $\Lambda_e$ are empty and Eq. (16) is nothing but the Schrödinger equation for the scattering region alone. Hence, once a candidate solution has been found one checks that

$$V P_{\text{sr}} \psi_{\text{sr}} - V \Phi_e q_e = 0 \tag{19}$$

to verify that the solution indeed satisfies the original set of equations.

Figure 2a shows an example of the eigenvalues of $H_{\text{eff}}(E)$ as a function of $E$ for an integrable billiard [the specific Hamiltonian is introduced later in Eq.(23)]. The vanishing eigenvalues correspond to possible bound states solutions, while the arrows indicate those that are true bound states.

## 3.4 Algorithm III: improved convergence thanks to gradient calculation

To improve the convergence of the root finder algorithm we expand $H_{\text{eff}}(E)$ up to a linear order in $E$. Since $H_{\text{eff}}(E)$ is a Hermitian matrix, we use first-order perturbation theory to obtain the gradient of its eigenvalues with respect to $E$,

$$\frac{d\varepsilon_\alpha}{dE} = \psi_{\text{eff},\alpha}^\dagger \frac{dH_{\text{eff}}}{dE} \psi_{\text{eff},\alpha}, \tag{20}$$

where $\varepsilon_\alpha$ is an eigenvalue of $H_{\text{eff}}$ and $\psi_{\text{eff},\alpha}$ the corresponding eigenvector. Knowing the gradient allows to use root-finding algorithms that converge much faster, such as the Newton-Raphson method.

When implementing this algorithm, one needs to evaluate $dH_{\text{eff}}/dE$ which amounts to calculating the derivatives $d\Lambda_e/dE$ and $d\Phi_e/dE$. These matrices are built from the non-Hermitian generalized eigenproblem (5). To compute these derivatives, we follow Ref. 28 for a general eigenproblem of the form

$$Ax = \kappa Bx, \tag{21}$$

where $A$ and $B$ are the matrices on the left and right-hand side of Eq. (5), $x = \begin{pmatrix} \phi \\ \xi \end{pmatrix}$ and $\kappa = 1/\lambda$. We assume the eigenvalue $\kappa$ to be non-degenerate. Since the matrix $B$ does not depend on energy, taking the first derivative of Eq. (21) gives

$$A\frac{dx}{dE} - \kappa B\frac{dx}{dE} - \frac{d\kappa}{dE}Bx = -\frac{dA}{dE}x.$$

The left-hand side of the above equation can be rewritten as an extended matrix-vector product of the form

$$\begin{pmatrix} A - \kappa B & -Bx \end{pmatrix} \begin{pmatrix} \frac{dx}{dE} \\ \frac{d\kappa}{dE} \end{pmatrix} = -\frac{dA}{dE}x. \tag{22}$$

This system cannot be solved directly since there are $2N_t + 1$ unknowns but only $2N_t$ equations. An additional constraint arises from the normalization of the eigenvector $x$. We choose to

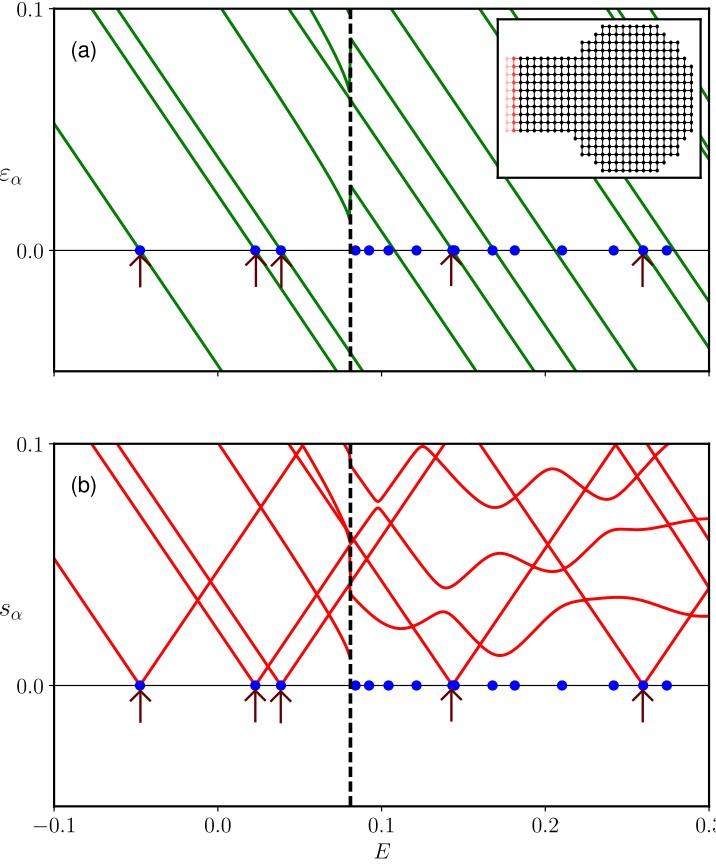

Figure 2: (a): eigenvalues $\varepsilon_\alpha(E)$ of Eq. (17) as a function of $E$ for a quantum billiard. (b): lowest singular values $s_\alpha(E)$ of the left-hand side of Eq. (11) for the same billiard. Inset: schematic of the billiard, described by Eq. (23) with $v_g = -0.1$. The blue dots correspond to the eigenenergies of a truncated system that consists of the billiard plus a finite fraction of the lead. The arrows indicate the positions of the actual bound states of the infinite system. The black dashed line corresponds to the opening of the first mode of the lead, which is marked by a discontinuity in the eigenvalues $\varepsilon_\alpha(E)$ and the singular values $s_\alpha(E)$.

set the value of the largest component of $x$ to unity: writing $m$ for the index of the largest component of $x$, we have $x_m \equiv 1$ therefore $\frac{dx_m}{dE} = 0$ and we can remove the corresponding $m$-th column from the left-hand side of Eq. (22). We are left with a system which, unless $\kappa$ corresponds to a degenerate eigenvector, has linearly independent columns [28] and can be solved numerically.

## 3.5 Visualization of algorithms I, II and III

Figure 2 shows the eigenvalues of Eq. (14) and the lowest singular values of Eq. (11) for an integrable quantum billiard (to be introduced in the next section) as a function of $E$. At energies below the band bottom, when there are no propagating modes (left of the vertical dashed line), we observe a one-to-one correspondence between eigenvalues going through

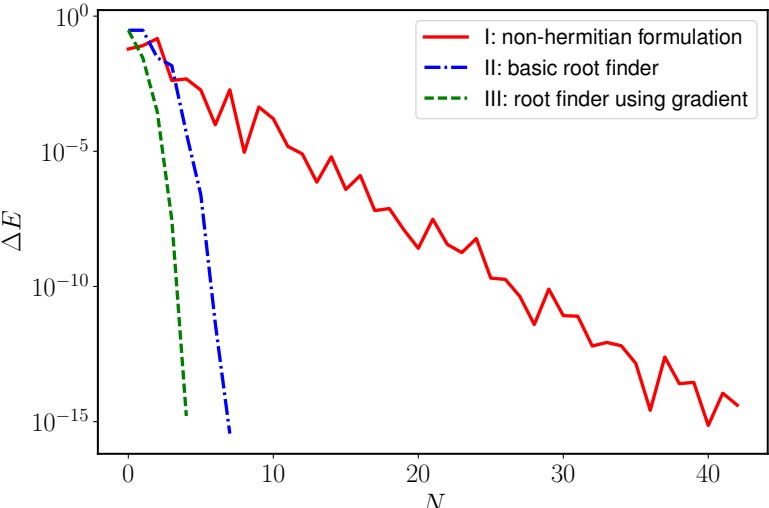

Figure 3: Convergence of the bound state energy as a function of the number of iterations for the three algorithms presented in the text.

zero, zero-valued minima of singular values, as well as eigenstates of a truncated system. Due to the finite size of the truncated system there is a small mismatch between its eigenenergies and the zero crossings/minima. For energies (right of the vertical dashed line), the scattering region is connected to a continuum of states. The transition is marked by a discontinuity in the eigenvalues of Eq. (14). In this regime there exist eigenstates of the truncated system that do not match a vanishing singular value. These states correspond to the continuum and more of them appear as more lead cells are included in the truncated system. The eigenvalues of Eq. (14) have more zeros than the singular values of Eq. (11) because some solutions of Eq. (14) are false positives that can be eliminated using Eq. (19).

The convergence rate of the different approaches is shown in Fig. 3, which confirms that algorithm III is the fastest. Since perfect convergence is achieved after only a few iterations, this algorithm is capable of obtaining the bound states for a cost comparable to that of calculating the propagating modes (scattering matrix).

## 4 Application to quantum billiards

We consider a circular (integrable) quantum billiard discretized on a square lattice with nearest neighbor hoppings:

$$H = 4t - t \sum_{\langle i,j \rangle} |i\rangle \langle j| + v_g \sum_{i \in \text{sr}} |i\rangle \langle i|, \quad t = 1. \tag{23}$$

Here, $\langle i, j \rangle$ stands for a summation over nearest neighbours. The summation in the last term is restricted to the scattering region, applying an onsite potential $v_g$ inside it, while this potential is equal to 0 in the lead.

We compute the density of states (DOS) of this system using Kwant [6], and the bound state spectrum using our algorithm. The results are shown in Fig. 4. We observe that there is an energy range in which bound states coexist with the continuum without mixing with it. At a certain energy a second propagating channel opens in the lead and washes out the remaining bound states. Below the bottom of the band, only bound states are present in the system. We emphasize that the bound states inside the continuum (BICs) present in the intermediate

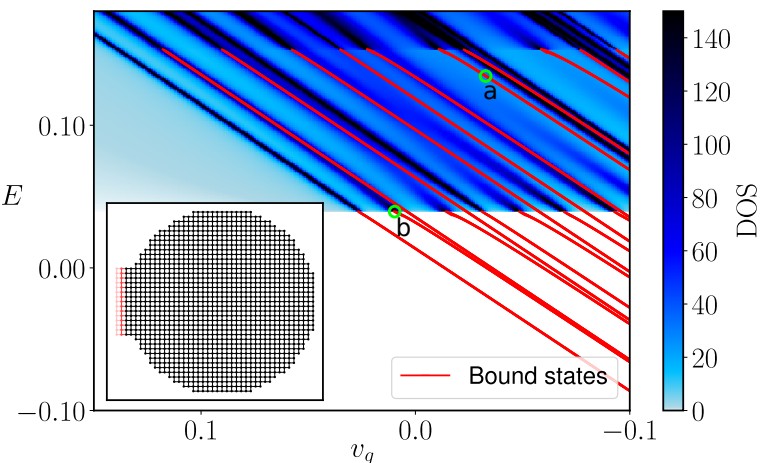

Figure 4: Color plot: density of states in the scattering region, from zero (white) to 150 (dark blue) as a function of $v_g$. Red lines: energies of bound states calculated using our algorithm. Inset: the scattering region (black) and a few cells of the lead (red). This system consists of about one thousand sites. The wavefunctions corresponding to the two circles (a) and (b) are shown in Figures 5a) and 5b).

energy range $0.03 \leq E \leq 0.15$ are very difficult to observe with a direct diagonalization of a finite system, since there is no simple way to distinguish eigenstates that originate from the continuum from actual bound states. Two examples of wavefunctions corresponding to particular bound states are shown in Fig. 5.

The presence of the BICs is a consequence of the symmetry of the circular billiard and of the corresponding lead: the lowest propagating mode is symmetric with respect to the $y$-axis while the BICs are antisymmetric, as shown in Fig. 5a), hence the absence of hybridization. On the other hand, the BICs do not appear in the chaotic billiard of Fig. 6 which does not have reflection symmetry.

# 5 Topological materials

We now present applications to topological materials where our algorithm is used to illustrate the bulk-boundary correspondence. In these three examples, we consider infinite systems of different dimensionality that are cut at $x = 0$ to form, respectively, a semi-infinite wire (with 0D Majorana), a half plane (with 1D edge state), and half space (with a 2D Fermi arc surface state). In all these examples we consider a lattice termination boundary with $H_{sr} = H$, and $P_{sr} = 1$.

## 5.1 One-dimensional superconducting nanowire, Majorana bound states

The starting point for the topological superconducting nanowire is the Bloch Hamiltonian from Ref. 12:

$$H(k) = \tau_z(\mu - 2t) + \sigma_z B + \tau_x \Delta + \tau_z t \cos(k) + \tau_z \sigma_x \alpha_{so} \sin(k), \tag{24}$$

where the Pauli matrices $(\tau_0, \tau_x, \tau_y, \tau_z)$ and $(\sigma_0, \sigma_x, \sigma_y, \sigma_z)$ act in the Nambu (electron-hole) space and spin space, respectively. The parameter $\Delta$ is the superconducting gap, $\mu$ the chemical potential, $\alpha_{so}$ the strength of the spin-orbit coupling and $B$ the Zeeman magnetic field.

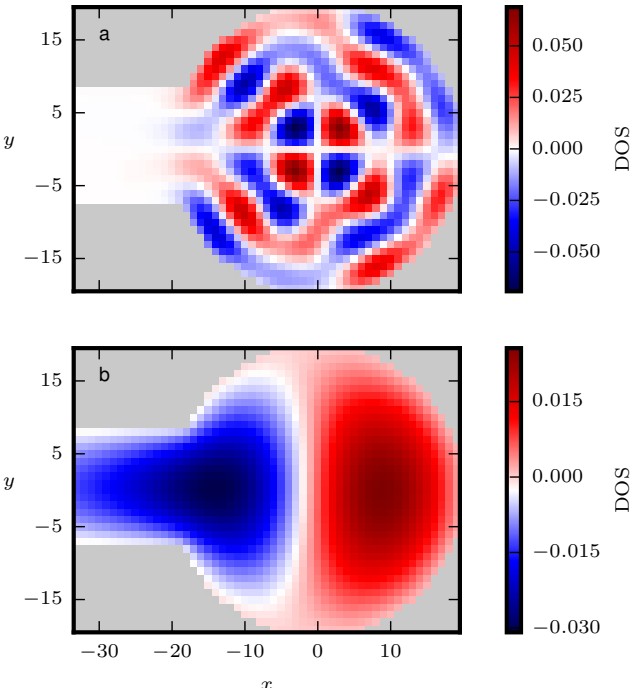

Figure 5: Real part of the wavefunction for two bound states of the billiard of Fig. 4. The parameters correspond to the white and bicolor circles (a) and (b) of Fig. 4. State (a) is a bound states that coexists with the continuum. State (b) was selected at an energy below (but very close to) the opening of the first mode. This way, the bound state decays only slowly in the lead and would be costly to compute by diagonalizing a truncated system.

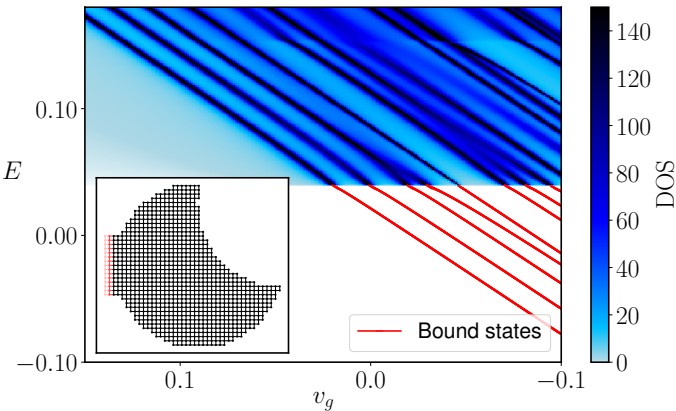

Figure 6: Same as Fig. 4, but with a chaotic scattering region.

Equation (24) corresponds to a tight-binding Hamiltonian with $N_\text{t} = 4$,

$$H = \tau_z(\mu - 2t) + \sigma_z B + \tau_x \Delta, \tag{25}$$
$$V = \tau_z t + i\tau_z \sigma_x \alpha_\text{so}. \tag{26}$$

In Fig. 7 we compare diagonalization of a finite system with the output of our algorithm. The upper panel shows the spectrum of a superconducting wire of finite length as a function of the chemical potential. The region with a dense set of energies corresponds to the continuum while

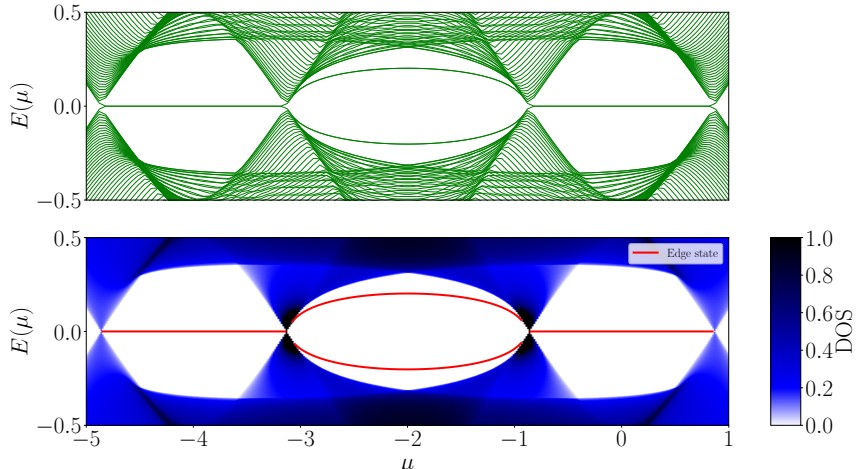

Figure 7: Spectrum of a superconducting nanowire as a function of chemical potential $\mu$. Upper panel, the eigenvalues of a finite wire of 100 sites obtained using direct diagonalization. Lower panel, color plot of the local DOS (blue) together with the bound states calculated directly in the thermodynamic limit (red lines). The Hamiltonian is given in Eq. (24) with $B = 1$, $t = -1$, $\alpha_{so} = 0.5$ and $\Delta = 0.5$.

the Majorana bound states are isolated. The lower panel displays the bound states as calculated with our algorithm and the DOS obtained with Kwant [6]. Both panels match perfectly, but the bound state algorithm has the advantage of working directly in the thermodynamic limit which enables it to distinguish bound states from the continuum.

In Fig. 8 we compute the wave function $\psi_{sr}^{\dagger}\psi_{sr}$ at the boundary of the topological region. Close to the quantum phase transition between the topological and non-topological phases the brute force diagonalization exhibits finite size corrections.

Another application of the bound state algorithm to the same Hamiltonian, illustrated in Fig. 9, is the calculation of the Andreev bound states of a Josephson junction. The hopping term between the left superconductor and the normal metal is described by Eqs. (25) and (26) but acquires a phase difference $\varphi$ such that

$$V_{SN} = \exp(i\varphi\tau_z)(\tau_z t + i\tau_z\sigma_x\alpha_{so}).  \tag{27}$$

The normal section in the center consists of one site where the gap $\Delta = 0$ and an onsite potential $V_N\tau_z$ is added. The Fig. 9 shows the energy of the Andreev bound states as a function of the phase difference $\varphi$ in the topological (a) and trivial (b) case. As expected, one recovers the $4\pi$ and $2\pi$ periodicity of the respective spectrum.

## 5.2 Quantum spin Hall effect

We continue with the two-dimensional BHZ model for the quantum spin Hall effect [29]:

$$H(\mathbf{k}) = \begin{pmatrix} h(\mathbf{k}) & 0 \\ 0 & h^*(-\mathbf{k}) \end{pmatrix},  \tag{28}$$

where

$$h(\mathbf{k}) = \epsilon(\mathbf{k}) + \vec{d}(\mathbf{k})\cdot\vec{\sigma},$$

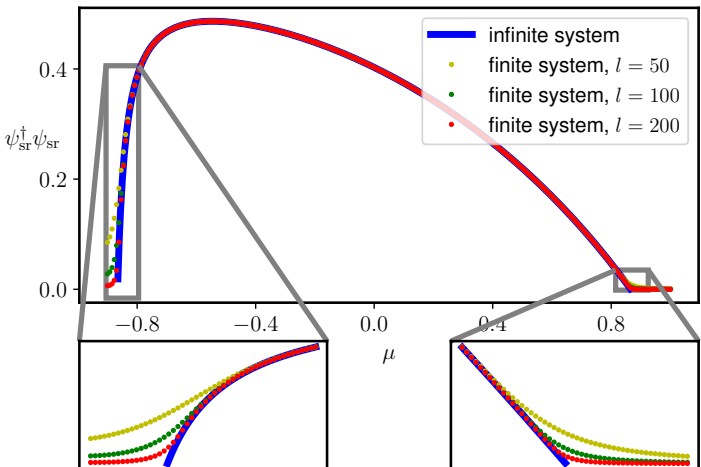

Figure 8: Weight of the Majorana bound state at the system boundary $\psi_{\text{sr}}^{\dagger}\psi_{\text{sr}}$ as a function of the chemical potential $\mu$. The solid line is computed using our algorithm, i.e., simulating a semi-infinite system, while the dotted lines correspond to the diagonalization of finite systems with sizes $l = 50, 100$ and $200$. The finite size effects are more pronounced close to the critical point where the gap closes, while our algorithm is only limited by machine precision. Insets: zoom of the main curve.

and

$$\begin{aligned}
\epsilon(\mathbf{k}) &= C - D(k_x^2 + k_y^2), \\
\vec{d}(\mathbf{k}) &= (Ak_x, -Ak_y, M - B(k_x^2 + k_y^2)).
\end{aligned}$$

The vector $\vec{\sigma} = (\sigma_x, \sigma_y, \sigma_z)$ contains the Pauli matrices. The constants $A$, $B$, $C$, $D$ and $M$ are material parameters. The tight-binding model is two-dimensional with the onsite Hamiltonian

$$H_0 = (C - 4D)\sigma_0 + (M - 4B)\sigma_z, \tag{29}$$

and hopping matrices

$$\begin{aligned}
V_x &= D\sigma_0 + B\sigma_z + \frac{1}{2i}A\sigma_x, \\
V_y &= D\sigma_0 + B\sigma_z - \frac{1}{2i}A\sigma_y
\end{aligned} \tag{30}$$

along the two directions.

We apply the Bloch theorem in the $x$-direction, and compute the bound state spectrum and the DOS as a function of $k_x$. We calculate the bound state of the effective quasi-one-dimensional system (parametrized by $k_x$) given by

$$\begin{aligned}
H &= H_0 + V_x e^{-ik_x} + V_x^{\dagger} e^{ik_x}, \\
V &= V_y,
\end{aligned} \tag{31}$$

for which our algorithm can be applied directly. The results are shown in Fig. 10 where the DOS in the topological insulating phase is shown together with the positions of the bound states. As expected, the helical edge states appear in the gap.

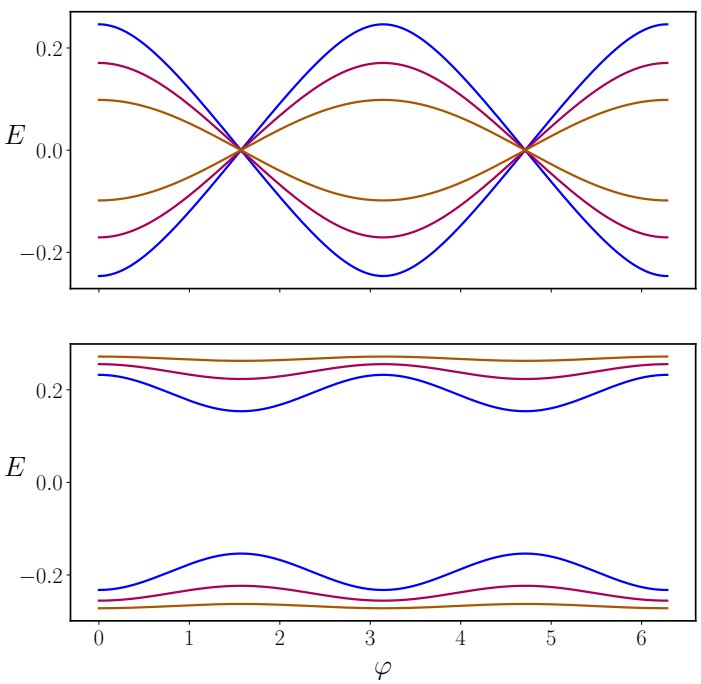

Figure 9: Andreev bound states in a Josephson junction as a function of the superconducting phase difference $\varphi$. Parameters are set to $\Delta = 0.5$, $\mu = 0.5$, $V_N = 1.25$ (blue), 2.5 (red) and 5 (orange), $\alpha_{so} = 0.5$. $B = 1$ in the topological regime (panel a) and $B = 0.4$ in the trivial one (panel b).

## 5.3 Fermi arcs in Weyl semimetals

We conclude with a last example that uses the same procedure for a three-dimensional Weyl semimetal whose Bloch Hamiltonian is given by [30]

$$H(\mathbf{k}) = \tau_z \left[ t\sigma_x \sin(k_x) + t\sigma_y \sin(k_y) + t_z \sigma_z \sin(k_z) \right] + \mu(\mathbf{k})\tau_x \sigma_0 + \frac{1}{2} b_0 \tau_z \sigma_0 + \frac{1}{2}\beta \tau_0 \sigma_z, \quad (32)$$

where

$$\mu(\mathbf{k}) = \mu_0 + t(2 - \cos(k_x) - \cos(k_y)) + t_z(1 - \cos(k_z)).$$

This Hamiltonian corresponds to the 3D tight-binding model

$$
\begin{aligned}
H_0 &= (\mu_0 + 2t + t_z)\tau_x + \frac{1}{2}b_0\tau_z + \frac{1}{2}\beta\sigma_z, \\
V_x &= \frac{1}{2}it\tau_z\sigma_x - \frac{1}{2}t\tau_x\sigma_0, \\
V_y &= \frac{1}{2}it\tau_z\sigma_y - \frac{1}{2}t\tau_x\sigma_0, \\
V_z &= \frac{1}{2}it_z\tau_z\sigma_z - \frac{1}{2}t_z\tau_x\sigma_0.
\end{aligned}
\quad (33)
$$

Once again, applying the Bloch theorem in the $x$-direction and using $k_y$ and $k_z$ as parameters, so that

$$
\begin{aligned}
H &= H_0 + (V_y e^{-ik_y} + V_z e^{-ik_z} + h.c.), \\
V &= V_x,
\end{aligned}
\quad (34)
$$

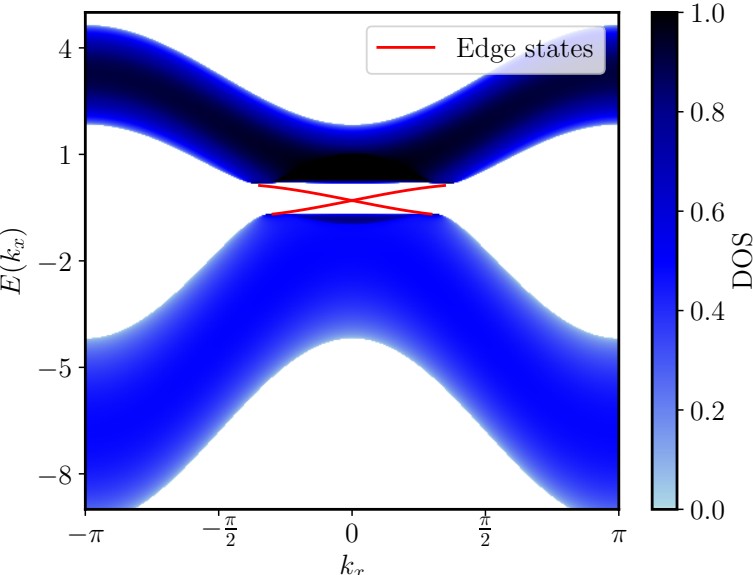

Figure 10: Color map of the density of states in the quantum spin Hall effect (blue) and corresponding edge states (red). The parameters have been fixed so that the model is in the topologically non-trivial insulating phase, $A = 0.5, B = 1, C = 0, D = 0.3, M = -1$.

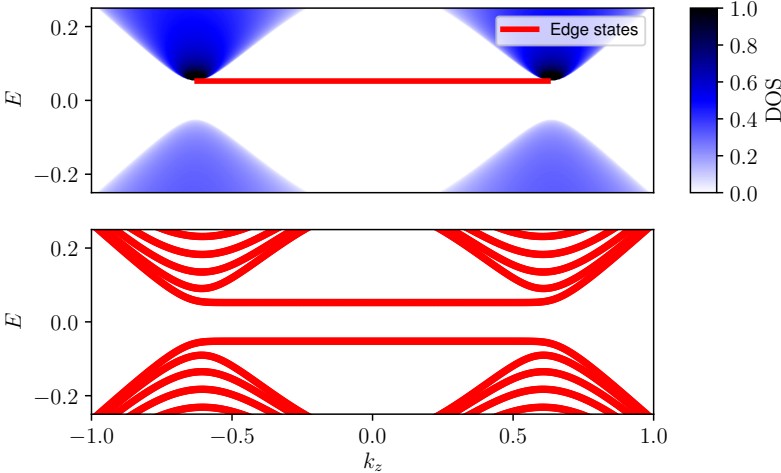

Figure 11: Upper panel: Color map of the DOS of the Weyl model and corresponding surface states as a function of energy $E$ and $k_z$ for a fixed value of $k_y = \pi/120$. Lower panel: Corresponding spectrum for a finite stack of 120 layers along the x direction. The parameters were set to $t = 2, t_z = 1, \mu = -0.3, b_0 = 0, \beta = 1.2$. A similar simulation is featured in Ref. 31.

which can be used directly in our algorithm. The results are shown in Fig. 11 and 12. As expected, we obtain the Fermi arcs that couple the two Weyl points. Again, our algorithm allows to study a single surface of the Weyl semimetal, in contrast with the finite size calculations (shown in the lower panel of Fig. 11) where states belonging to the two surfaces hybridize.

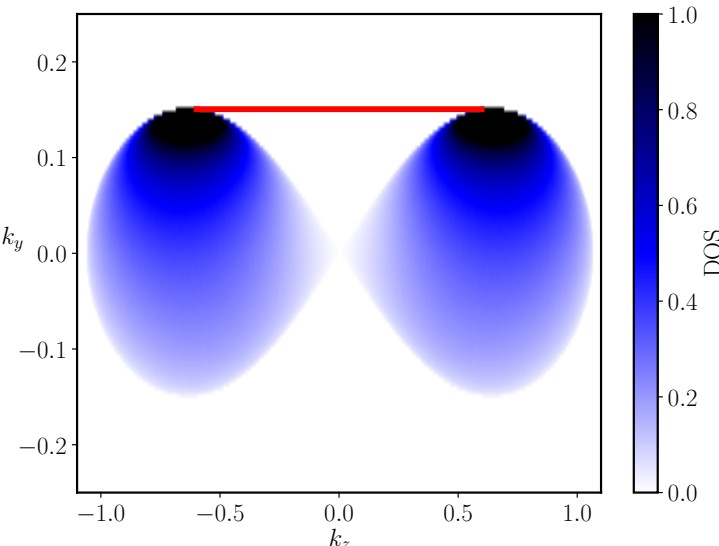

Figure 12: Same as the upper panel of Fig. 11 with an alternative view: the energy $E = 0.3$ is fixed and the DOS is plotted as a function of $(k_y, k_z)$.

## 6 Conclusion

We have derived a robust and efficient algorithm to compute the energies and the wavefunctions of the bound states of infinite quasi one-dimensional systems described by general tight-binding Hamiltonians. We have applied our approach to a variety of systems and shown that it can efficiently calculate the bound states, including the situations where other approaches would fail or become computationally prohibitive.

This algorithm can be used on its own to obtain e.g. the Josephson relation of super-conducting-normal-superconducting junctions, the properties of quantum wells or characterize topological materials and their interfaces. It may also be used to obtain the properties of perfect surfaces or interfaces (in arbitrary topological, metallic or superconducting materials) that can then be used as the starting point for further calculations.

## Acknowledgments

We thank Daniel Varjas for pointing out the need for orthogonalization of the evanescent modes in presence of additional degeneracies.

**Funding information** Financial support from US Office of Naval Research (ONR) is gratefully acknowledged. AA and MW were supported by the Netherlands Organisation for Scientific Research (NWO/OCW). AA was also supported by the ERC Starting Grant 638760.

## A  Normalization of the bound state

Bound states should be correctly normalized:

$$\psi_{\text{sr}}^{\dagger}\psi_{\text{sr}} + \sum_{j\geq 1}\psi_{\alpha}(j)^{\dagger}\psi_{\alpha}(j) = 1. \tag{35}$$

However, our algorithm does not ensure this normalization automatically. For a given set $(\psi_{\text{sr}}, q_{e,\alpha})$ we find that

$$\psi_{\text{sr}}^{\dagger}\psi_{\text{sr}} + \sum_{j\geq 1}\psi_{\alpha}(j)^{\dagger}\psi_{\alpha}(j) \;=\; \psi_{\text{sr}}^{\dagger}\psi_{\text{sr}} + q_{e,\alpha}^{\dagger}\bigg[\sum_{j\geq 1}(\Lambda_e^{\dagger})^j\Phi_e^{\dagger}\Phi_e(\Lambda_e)^j\bigg]q_{e,\alpha}. \tag{36}$$

We recognize a geometric series and arrive at

$$\psi_{\text{sr}}^{\dagger}\psi_{\text{sr}} + \sum_{j\geq 1}\psi_{\alpha}(j)^{\dagger}\psi_{\alpha}(j) = \psi_{\text{sr}}^{\dagger}\psi_{\text{sr}} + q_{e,\alpha}^{\dagger}Nq_{e,\alpha}, \tag{37}$$

with the matrix $N$ defined as

$$N_{mn} \equiv \frac{1}{1-\lambda_n\lambda_m^*}\lambda_n\lambda_m^*\big(\Phi_e^{\dagger}\Phi_e\big)_{m,n}. \tag{38}$$

Eq. (37) can now be used to normalize the bound state wave functions to unity.

## B  Proof of Eq. (13)

To prove Eq. (13), we begin by multiplying Eq. (6) by $\Phi_e^{\dagger}$:

$$\Phi_e^{\dagger}V\Phi_e + \Phi_e^{\dagger}(H-E)\Phi_e\Lambda_e + \Phi_e^{\dagger}V^{\dagger}\Phi_e(\Lambda_e)^2 = 0. \tag{39}$$

The complex conjugate of the above equation reads

$$(\Lambda_e^*)^2\Phi_e^{\dagger}V\Phi_e + \Lambda_e^*\Phi_e^{\dagger}(H-E)\Phi_e + \Phi_e^{\dagger}V^{\dagger}\Phi_e = 0. \tag{40}$$

Now, multiplying Eq. (39) by $\Lambda_e^*$ on the left and Eq. (40) by $\Lambda_e$ on the right, we arrive after substracting one equation from the other at

$$\big[\lambda_\alpha^* - (\lambda_\alpha^*)^2\lambda_\beta\big]\,\Phi_e^{\dagger}V\Phi_e\big|_{\alpha\beta} = \big[-\lambda_\alpha^*(\lambda_\beta)^2 + \lambda_\beta\big]\,\Phi_e^{\dagger}V^{\dagger}\Phi_e\big|_{\alpha\beta}. \tag{41}$$

Since we are dealing with evanescent states, we can simplify by $1-\lambda_\alpha^*\lambda_\beta \neq 0$ and arrive at

$$\lambda_\alpha^*\,\Phi_e^{\dagger}V\Phi_e\big|_{\alpha\beta} = \lambda_\beta\,\Phi_e^{\dagger}V^{\dagger}\Phi_e\big|_{\alpha\beta} \tag{42}$$

which is essentially Eq. (13).

## C  Degenerate eigenvalues

In some cases, the solutions $\lambda$ of Eq. (3) can be degenerate, as in the quantum spin Hall model of Sec. (5.2) where the degeneracies arise because of the two species of spin. A set of degenerate eigenvectors is not uniquely defined, as any linear combination is still a valid solution of the eigenproblem (3). This means that the matrix $\Phi_e$ that was introduced in Sec. 2.3

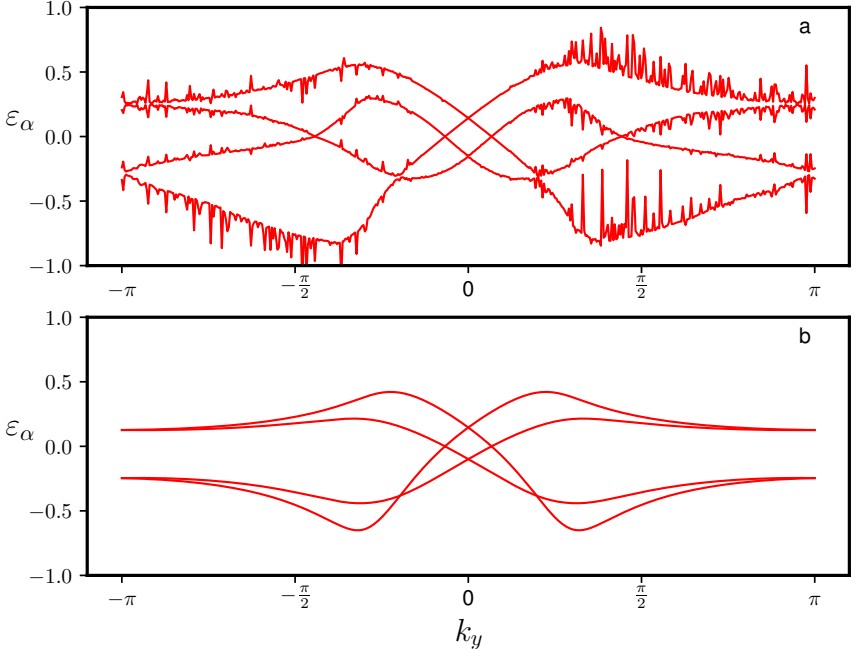

Figure 13: Eigenvalues of $H_{\mathrm{eff}}(E, k)$ for the quantum spin Hall model (Sec. 5.2) as a function of the momentum, $E = -0.4$. In (a) the vectors $\Phi_e$ are normed but not orthogonalized, in (b) the vectors are orthogonalized using a QR decomposition.

can be replaced by $\Phi'_e \equiv \Phi_e T$, where $T$ is an invertible matrix equal to unity except in the block corresponding to degenerate eigenvalues. This modification results in an uncertainty of the l.h.s. of Eq. (14) since

$$
H'_{\mathrm{eff}}(E) \equiv \begin{pmatrix} H_{\mathrm{sr}} - E & P_{\mathrm{sr}}^T V^\dagger \tilde{\Phi}'_e \tilde{\Lambda}_e \\ \tilde{\Lambda}_e^* \tilde{\Phi}_e'^\dagger V P_{\mathrm{sr}} & -\tilde{\Lambda}_e^* \tilde{\Phi}_e'^\dagger V \tilde{\Phi}'_e \end{pmatrix} \tag{43}
$$

$$
= \begin{pmatrix} 1 & 0 \\ 0 & T^\dagger \end{pmatrix} \begin{pmatrix} H_{\mathrm{sr}} - E & P_{\mathrm{sr}}^T V^\dagger \tilde{\Phi}_e \tilde{\Lambda}_e \\ \tilde{\Lambda}_e^* \tilde{\Phi}_e^\dagger V P_{\mathrm{sr}} & -\tilde{\Lambda}_e^* \tilde{\Phi}_e^\dagger V \tilde{\Phi}_e \end{pmatrix} \begin{pmatrix} 1 & 0 \\ 0 & T \end{pmatrix} \tag{44}
$$

does not have the same eigenvalues as $H_{\mathrm{eff}}$, unless $T$ is unitary. (We use the fact that $\Lambda_e$ commutes with $T$.) This leads to a problem during the root-finding phase of the algorithms of Sec. 3.1 and 3.3: in a naive implementation $H_{\mathrm{eff}}$ is evaluated multiple times for different $E$, each time with an effectively random $T$. The resulting fluctuations, shown in Fig. 13(a), are incompatible with efficient root-finder routines. The algorithm presented in 3.4 is not considered here as the derivatives computed in that section are not well defined for degenerate eigenvalues.

   It is important to note that the transformation from $H_{\mathrm{eff}}(E)$ to $H'_{\mathrm{eff}}(E)$ does not affect the zero-valued eigenvalues – the only ones with a physical meaning. Furthermore, replacing $\Phi$ by $\Phi T$ also changes $q_e$ by $T^{-1} q_e$, so that the final wavefunction $\Psi(j)$ in Eq. (7) remains unchanged. This is why the algorithms of Sec. 3 remain correct, but require a fluctuation-tolerant root-finder when implemented naively. In practice, it is preferable to fix a unique spectrum such that an efficient standard root-finder can be used. The matrices $\Phi_e$ cannot be fixed in a unique way, but one can choose the column vectors such that the degenerate ones are orthogonal. To this end, one performs the QR decomposition $\Phi_e = QR$, where $Q$ is an



orthogonal matrix and $R$ an upper triangular matrix, and substitutes in Eq. (17):

$$\Phi_e \rightarrow Q, \tag{45}$$
$$\Lambda_e \rightarrow R\Lambda_e R^{-1}, \tag{46}$$
$$q_e \rightarrow R^{-1}q_e. \tag{47}$$

The resulting smooth eigenvalues are shown in Fig. 13(b).

Orthogonalizing $\Phi_e$ forces the matrix $T$ to be unitary, as we can understand using the following geometrical argument. Any superposition on two vectors $\mathbf{e}_1$ and $\mathbf{e}_2$ of the $(x, y)$ plane is still a valid basis as long as they are not collinear. If one forces the vectors to be perpendicular to each other, then the only transformation left is a rotation, in other words a unitary transformation.

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
