# Peer review of "A general algorithm for computing bound states in infinite tight-binding systems"

_SciPost Physics, doi:SciPost Phys. 4, 026 (2018)_

## Round 1 · Referee Report · Anonymous (Referee 1) · 2018-1-3

Strengths

1- The paper provides a systematic formulation of the bound-state problem using self energies representing asymptotic scattering regions. 2- The paper describes and compares numerical alogrithms to evaluates the ensuing accuracy and convergence properties. 3- The paper applies these algorithms to a varity of relevant and topical model scenarios.

Weaknesses

1- The paper could do a better job to put this approach in the context of previous works.

Report

This is a useful paper for researchers in a variety of fields that concern single-particle physics, in particular in low-dimensional systems. I congratulate the authors for the careful exposition of their ideas.

It should be noted that the general idea of including the leads via the surface green function has been mentioned before also for the context of bound states. See e.g. Appendix D in PHYSICAL REVIEW B, VOLUME 63, 245407 (where this problem was then classified as "difficult".) The bound states are also implicit in J. Appl. Phys. 81, 7845 (1997) (see discussion of below eq 39 and also remark below eq (1) in https://www.wsi.tum.de/Portals/0/Media/Publications/047559a5-94fa-4aa7-9bb9-f9b09fd17a86/CompPhys2010.pdf)
Note that this is in the more challenging context of selfconsistent NEGF calculations. I identified these works in a brief search and would not be surprised if you could find more papers with this general idea. On the other hand, I am not aware of an implementation as described here (however, the authors should make an effort to verify this in the view of the mentioned works). I also miss a discussion that set these ideas into the broader context of scattering quantization and other alternatives - see requested changes.

Requested changes

1- Discuss predecessors of the idea, such as PHYSICAL REVIEW B, VOLUME 63, 245407, J. Appl. Phys. 81, 7845 (1997) or other works that may pursue this line of thought even more closely.

2- Also expand the description of general context with reference to earlier scattering approaches such as by Smilanksy, Bogomolny et al. For a useful guide see https://arxiv.org/abs/nlin/0204061v1 (and therein eg refs [46-49],[72]).

3- Also worth mentioning is the boundary integral method, e.g. for open systems: Boundary element method for resonances in dielectric microcavities: J Wiersig, Journal of Optics A: Pure and Applied Optics 5 (1), 53 (there applied to resonances - but these become bound states when the interior confines such states).

---

## Round 1 · Referee Report · Anonymous (Referee 2) · 2018-2-9

Strengths

1- Presents numerical method for a broad and timely class of problems, namely finding the bound states in an infinite system with a scattering region. 2- Three methods are compared to evaluate efficiency and robustness. 3- Examples of the method are given applied to problems of current interest.

Weaknesses

1- The main weakness is that no new physics is included in the examples. The focus is technical, on the method itself.

Report

This manuscript presents a numerical method to find bound states in an infinite system consisting of a scattering region attached to leads-- the geometry of a quantum transport experiment, for instance. The authors find a formulation in which the bound states are given by the kernel of a Hermitian operator. They are then able to find efficient methods to tackle this problem and illustrate the workings of these methods. The paper closes with three examples of use of the method.

I think that this is a fine paper which may well become a standard reference on how to find bound states in such systems. As the authors' examples illustrate there are very many problems of current interest to which this can be applied. I expect the paper to be eagerly received by the community and put to immediate use.

A weakness of the paper is that no new physics is discussed. The examples that the authors treat are all already in the literature and their physics is well understood. These examples serve the important purpose of validation and illustration of the method, but it is somewhat disappointing that the authors did not include some physics that is at least somewhat new.

The paper itself is very well written and clear. I just have a few small points that require changes (below).

Requested changes

  1. The explicit energy dependency should be given in one of the equations showing the system that has to be solved, ie. (12), (14), or (17). The reader should be able to tell at a glance that a nonlinear eigenproblem is involved. I suggest adding the energy dependence to (14), the main result of the paper.

  2. There's a problem just after Eq (11) referring to Eq (14). The truncated quantities should be defined in connection with (11) since it comes first.

  3. Clarification of the Hamiltonian (23) is needed. The authors use the term "billiard", which means that the onsite potential is either 0 or infinity, yet then refer to a finite value of v_g (top of p. 12). I conclude that the authors are using a frame of tight-binding sites in which a large but finite v_g is then used to sculpt out the potential they want. If this is the case, an explicit statement to this effect should be added. (If not, explain...)

  4. The green circles in Fig 4 are essentially invisible when printed. A change in color scheme is needed.

---

## Editorial Decision

published